# Fault Early Warning Model for High-Speed Railway Train Based on Feature Contribution and Causal Inference

**DOI:** 10.3390/s22239184

**Published:** 2022-11-25

**Authors:** Dian Liu, Yong Qin, Yiying Zhao, Weijun Yang, Haijun Hu, Ning Yang, Bing Liu

**Affiliations:** 1State Key Lab of Rail Traffic Control & Safety, Beijing Jiaotong University, Beijing 100044, China; 2Locomotive & Car Research Institute, China Academy of Railway Sciences Group Co., Ltd., Beijing 100081, China; 3China Railway Engineering Design and Consulting Group Co., Ltd., Beijing 100055, China

**Keywords:** causal inference, SHAP, interpretability, data-driven, fault warning

## Abstract

The demands for model accuracy and computing efficiency in fault warning scenarios are increasing as high-speed railway train technology continues to advance. The black box model is difficult to interpret, making it impossible for this technology to be widely adopted in the railway industry, which has strict safety regulations. This paper proposes a fault early warning machine learning model based on feature contribution and causal inference. First, the contributions of the features are calculated through the Shapley additive explanations model. Then, causal relationships are discovered through causal inference models. Finally, data from causal and high-contribution time series are applied to the model. Ablation tests are conducted with the Naïve Bayes, Gradient Boosting Decision Tree, eXtreme Gradient Boosting, and other models in order to confirm the efficiency of the method based on early warning data regarding the on-site high-speed train traction equipment circuit board failure. The findings indicate that the strategy improves the evaluation markers, including the early warning accuracy, precision, recall, and F1 score, by an average of more than 10%. There is a 35% improvement in the computing efficiency, and the model can provide feature causal graph verification for expert product decision-making.

## 1. Introduction

According to the 2021 Statistical Bulletin of the China National Railway Group Co., Ltd., the operating mileage of the entire railway is 150,000 km, and the number of passengers is 2.533 billion, an increase of 366 million people compared to the previous year. The Bulletin also shows that there are 40,000 km of high-speed rail and 33,221 EMUs. Based on the on-board sensors, ground sensors, inspection and maintenance equipment, and experimental and testing equipment, China’s railways have accumulated massive amounts of structured, semi-structured, unstructured, and streaming data on the health status of key equipment. Additionally, China has carried out the development of PHM-oriented high-speed train pedigree product technology platforms [1]. The data has significant mining and application value, and data-driven fault diagnosis has been successfully applied in many fields, such as the field of disease medical treatment [2], the field of financial credit [3], and the field of energy [4] and electricity [5,6]. In the above scenarios, theoretical principles, historical behaviors, and abnormal influences are deposited in the characteristic data, and people can use a large amount of prior and already diagnosed data to carry out data-driven fault diagnosis and early warning.

There are some research gaps in previous work that need to be addressed. In the early days, fault diagnosis expert systems were mostly based on physical laws, rules, and cases [7,8]. The diagnostic rules of the model are constructed by product experts based on their knowledge and experience. The advantage is that they are less dependent on the data and can be interpreted. It is feasible to use these methods when the physical model of the research object is relatively simple and the feature dimension is low. When the physical model of the research object has complex feature relationships, it is often difficult for experts to build a physical model through experience.

With the development of data, algorithms, and computing power, some machine learning algorithms have been applied to fault diagnosis systems, including KNN [9], GBDT [10], and XGBoost [11]. These machine learning models learn diagnostic knowledge from data. However, the experience and knowledge of industry experts are rarely involved in the construction of algorithms and models. This makes it difficult to reflect the contributions of features and the causal relationships between features and faults in the model. Introducing irrelevant features into the model will reduce the efficiency of the model and, even worse, it will affect the accuracy of the model when it contains confusing features.

In the field of the fault diagnosis of key equipment for high-speed railway trains, the method of feature recognition [12,13,14] and the method of deep learning [15,16] have achieved good results. However, the fault diagnosis of this “black box model” is limited by the model principle, data magnitude, and parameter complexity. This makes it difficult for the models to effectively use expert knowledge and experience. It is unable to meet the requirements regarding the interpretability of field products. After the failure occurs, the cause analysis and technical improvement have an important impact on the improvement of the product performance and the safety of rail transit operations. Due to the lack of causal inference tools, experts rely on industry experience to replace parts individually and test them frequently, which usually requires a great deal of time and costs. Even when multi-part combined features cause failures, these failures can be difficult to reproduce, thereby leading to the failure to improve the product.

Modeling that relies solely on statistical correlations derived from the data has limitations. It is difficult to use for decision making due to its lack of causal considerations. Firstly, machine learning models that use correlations have a poor generalization ability and stability and are highly susceptible to scene changes or outliers in the dataset. Furthermore, a machine learning model that relies too heavily on data fitting is similar to a black box and lacks interpretability. The stability and interpretability of causality can offer people enough confidence to make safe, scientific decisions. Product experts increase efficiency, reduce costs, and prevent losses when making product improvement decisions.

The main aim of the work and its conclusions are as follows. For nearly a century, scientists have believed in the statistical assertion that “correlation does not equal causation”, and traditional education in statistics has gradually formed a situation where it is taboo to discuss causality. This continued until the Turing Award winner Judea Pearl [17], the “father of Bayesian networks”, who developed machines to perform probabilistic reasoning, broke the taboo. He sharply criticized AI for being trapped in the quagmire of probabilistic associations and dispended with probabilistic inference in favor of causal inference theory. He believed that the breakthrough required to achieve strong AI lies in causal inference theory. In 2021, the Royal Swedish Academy of Sciences awarded the Nobel Prize in Economics to three scientists who have made outstanding contributions to the theory of causal inference. Among them, the research of Joshua D. Angrist and Guido W. Imbens showed that precise conclusions about causality can be drawn from data, helping to promote the use of causality in nature research. The high-speed railway fault early warning model based on feature contribution and causal inference can improve the operating efficiency and accuracy of the model. Interpretable causal model diagrams can more effectively utilize the industry knowledge and experience of experts. At the same time, it provides experts with verification of, and an intuitive reference for, the causal relationships of features. It can provide decision-making data support for experts to improve products and ultimately improve the efficiency of the analysis of failures and technical improvements.

This paper takes the electronic board failure warning of the traction converter, the key equipment of the high-speed electric multiple units (EMU), as the application scenario and research object. Firstly, based on the feature contribution of the Shapley additive explanations (SHAP) model, the causal features that may lead to the failure are preliminarily screened out from a large number of features. Then, we draw a causal inference model diagram based on expert knowledge and experience to describe the possible causal network relationship between each feature and fault. Additionally, we verify the accuracy of the causal network through the steps of identification, estimation, and refutation of the DoWhy model [18,19]. Finally, the historical time series data on the causal characteristics are introduced to improve the accuracy and operation efficiency of the machine learning model. The validity of the model is verified by an ablation experiment with high-speed railway field application data. It provides a more accurate fault early warning model for product safety application and provides a causal inference decision-making reference for product experts aiming to optimize products.

Our main contributions are summarized below:We combine the causal inference method used in the field of economics with the method of machine learning in the AI field and use this method to solve the problem of the early warning of key equipment failures in high-speed railway trains.By introducing characteristic time series data with significant contribution and causal relationships, the prediction accuracy of the classic machine learning model is improved, and the computational complexity is reduced.We propose a set of process methods for the application of causal inference to product experts so as to analyze the causes of product failures.Based on the real scenario of high-speed railway on-site operation, a set of methods in which product experts and data experts contribute their respective skills and cooperate to solve practical problems are proposed.

The rest of the paper is organized as follows: Section 2 presents the research methodology, model, and procedures, and Section 3 details the dataset and experimental results. Section 4 discusses the validity of the results, industry application, shortcomings, and prospects for future work. Section 5 summarizes the conclusions.

## 2. Methods

This paper proposes a high-speed railway train fault early warning machine learning model based on feature contribution and causal inference. As shown in Figure 1, the method has four main steps: data preparation, feature contribution calculation, causal inference, and model training and prediction. Usually, the predictions provided by data experts through model algorithms and data iterations are difficult to explain to product experts, and the prior knowledge of product experts based on physical laws is also difficult to directly apply to model algorithms. The method proposed in this paper can combine the expertise of data experts and product experts. In the data preparation stage, product experts select the relevant feature ranges through their prior knowledge of the products. This can reduce the feature dimension of the model calculation. In the feature contribution calculation stage, data experts visualize the contributions of related features through the SHAP model, focusing on the features with high contributions. In the causal inference stage, product experts draw causal diagrams based on high-contribution features and their prior knowledge of the products. Product experts do not have to provide a complete causal diagram. Causal inference can also provide decision support. In the model training and prediction stage, the model efficiency is improved through feature dimensionality reduction in the pre-order stage, especially after the introduction of dependent variable time series data. Because the influence of confounding features is reduced, the model prediction can achieve better results.

### 2.1. Data Preparation

The data preparation section describes the types of sensor information, positive and negative sample balance processing, missing value processing, abnormal sample elimination, feature value normalization processing, and feature selection based on prior knowledge.

#### 2.1.1. Sensor Data Collection

More than 2000 sensor measuring points are distributed on high-speed railway trains in more than ten subsystems and different carriages. The information collected by each sensor is collected as a dataset through remote wireless transmission and offline copying. Each data sample contains features such as the time (YYYY-MM-DD hh-mm-ss), line, vehicle number, speed, temperature information, pressure information, voltage information, current information, status information, and sample label. An alluvial diagram of the sensor data type distribution is shown in Figure 2, and an example of the location of each sensor on the high-speed train is shown in Figure 3. The dataset template is shown in Table 1.

#### 2.1.2. Positive and Negative Sample Balance Processing

In the actual operation scenario of high-speed railway trains, the fault samples are different from the number of data samples in normal operating conditions by orders of magnitude. Most of the data are collected by the sensor is under normal working conditions. At this time, the train has no fault warning or alarm. If the training set is formed by random sampling in the total sample set, it will seriously affect the effect of the supervised model, such as the Naive Bayes classifier. For the research object, this paper carries out the balance processing of the positive and negative samples and maintains the positive samples and negative samples in the same order of magnitude.

#### 2.1.3. Missing Value Handling

As they are affected by model differences, operating conditions, and network transmission, the collected samples have missing data. Through the prior knowledge and data analysis of experts, the failure of the research object is not cumulative, and the failure of the research object may lead to missing eigenvalue sampling. Therefore, this paper does not adopt the method of supplementing the mean value or supplementing a valid value when dealing with missing values and adopts the method of uniformly replacing the missing values with “−1”.

#### 2.1.4. Abnormal Sample Removal

In this paper, samples with a data integrity below 80% are regarded as abnormal samples, and abnormal samples are excluded from the dataset. The reason why the completeness threshold is set at 80% is because it is a trade-off between sample richness and model accuracy. A threshold value that is too high, such as 95%, will cause a large number of samples to be excluded from the dataset, which reduces the sample richness. If the threshold is too low, such as 70%, the strategy of filling the missing feature data with “−1”, in this paper, will lead to a decrease in the model accuracy. Based on the iterative verification of the data and model in this paper, 80% is suitable. At the same time, abnormal samples with unreasonable sampling results are further eliminated through industry experience and prior knowledge.

#### 2.1.5. Normalization Processing

The relevant characteristics of the research object include physical quantities, such as the temperature, voltage, current, pressure, mass, and length. It is necessary to avoid the influences of different physical quantities and unit conversions. That is, features with large values have a negative effect on the model training. In this paper, the 0-mean normalization method (Z-score normalization) is used to preprocess the datasets involved in the training and validation. The dimensional expression is changed into a dimensionless expression, so that the mean value of each feature is 0 and the variance is 1, so as to avoid the dimensional differences between different physical features leading to the reduction in the model training effect and convergence speed. It shown in Expression 1:(1)$$Z=\frac{x-\frac{1}{N}\sum_{i=1}^{N}x_{i}}{\sqrt{\frac{1}{N}\sum_{i=1}^{N}{\left(x_{i}-\bar{x}\right)}^{2}}}$$
where *x* is the feature value of the current sample, and $\bar{x}$ is the average value of the dataset sample.

#### 2.1.6. Selection by Prior Knowledge

The dataset in this paper can be divided into categorical data, ordinal data, interval data, and ratio data according to the nature of the values, as shown in Table 2. The qualitative data is called qualitative data, and the latter three types of data are called quantitative data. According to the data types with different characteristics, combined with the experience of industry experts and prior knowledge, the preliminary screening of the characteristic values is carried out to reduce the computational dimension and difficulty of the analysis of the subsequent steps.

### 2.2. Feature Contribution Calculation

Machine learning has achieved remarkable results in the medical, financial, security, and other fields. However, the lack of interpretability severely limits its use in real-world tasks, especially in rail transit safety application scenarios [20]. In the field of rail transit, involving the safety of passenger life and property, understanding the reasons for which a model makes a specific prediction is often as important as the accuracy of the prediction. Models must help decision makers to understand how to use them properly. At the same time, the more severe the scenarios are, the more the models must provide evidence to show how they work and avoid mistakes.

Commonly used interpretability methods in industry practice include decision tree visualization and data 3D waterfall charts, as shown in Figure 4. Decision tree visualization can explain the decision-making process that the data pass through at each node and finally forms a decision, but the complex and large tree structure is difficult to read and understand. Meanwhile, 3D drawing can illustrate the status of each feature for its further analysis by experts, but it cannot reflect the relationships between features and the relationships between features and results.

With the development of the data volume, algorithms, and computing power, ensemble learning or deep learning models can achieve ideal results. However, the results predicted by complex models and large datasets are difficult to explain. BERT [21] from the Google team and GPT-3 [22] from the OpenAI team are two examples. There are 175 billion parameters and a 96-layer network in GPT-3. Faced with such a large model, even if experts can interpret it, the magnitude of the parameters of the model and the magnitude of the network layers make the interpretation difficult to visualize and understand.

The above situation creates a contradiction between the prediction accuracy and interpretability. Faced with this problem, the SHAP (Shapley additive explanations) [23] model proposed by Scott M. Lundberg can be used in engineering practice. The method is based on the cooperative game solution proposed by the Nobel laureate in economics, Lloyd S. Shapley [24]. By attributing the output value to the Shapley value of each feature, the contribution of each feature in the model to the result is evaluated.

Suppose the *i*-th sample is $x_{i}$ and the *j*-th feature of the i-th sample is $x_{ij}$. The predicted value of the model for this sample is $y_{i}$. The baseline of the entire model, such as the mean of the target variable across all samples, is $y_{base}$. Then, the SHAP value obeys the following equation:(2)$$y_{i}=y_{base}+f\left(x_{i1}\right)+f\left(x_{i2}\right)+\cdots +f\left(x_{ij}\right),$$
where $f\left(x_{ij}\right)$ is the SHAP value of $x_{ij}$. $f\left(x_{i1}\right)$ is the contribution of the first feature in the i-th sample to the final predicted value $y_{i}$. When $f\left(x_{i1}\right)$ > 0, this means that the feature improves the predicted value and also has a positive effect. On the contrary, it is shown that this feature reduces the predicted value, which is counterproductive.

Traditional feature importance analysis can quantify the importance of features, but it cannot explain how the features affect the prediction results. The greatest advantage of the SHAP value is that SHAP can not only quantify the influences of features in the sample but also show the positive and negative results of the influence. Its expression is:(3)$$g\left(z^{\prime }\right)=\emptyset_{0}+\sum_{j=1}^{M}\emptyset_{j},$$
where *M* is the number of input features, ∅ is the Shapley value for each feature, and $\emptyset_{0}$ is a bias term.

In the above formula, the key is to obtain $\emptyset_{j}$. For a certain feature *j*, Shapley values need to be calculated for all the possible feature combinations and their order and then weighted and summed. The expression is:(4)$$\phi_{j}=\underset{S\subseteq F\backslash \left\{j\right\}}{\sum }\frac{\left|S\right|!\left(\left|F\right|-\left|S\right|-1\right)!}{\left|F\right|!}\left[f_{S\cup \left\{j\right\}}\left(x_{S\cup \left\{j\right\}}\right)-f_{S}\left(x_{S}\right)\right].$$

In the expression, *F* is the set of all the features, $f_{S\cup \left\{j\right\}}$ is the model for training the *j*-th feature, $f_{S}$ is the model for training all the features, and $f_{S\cup \left\{j\right\}}\left(x_{S\cup \left\{j\right\}}\right)-f_{S}\left(x_{S}\right)$ is the difference between the model output of feature *j* and the model output of all features. $x_{S}$ represents the value of the input feature in the set *S*. Since the contribution of feature *j* depends on other features in the model, compute $S\subseteq F\backslash \left\{j\right\}$ for all the possible subsets of $f_{S\cup \left\{j\right\}}\left(x_{S\cup \left\{j\right\}}\right)-f_{S}\left(x_{S}\right)$. The weight in the formula can be interpreted as the total of the *F* features. Then, considering the order, these *F* features have a total of *F*! kinds of combinations. If a feature *j* is fixed, then there are $\left|S\right|!\left(\left|F\right|-\left|S\right|-1\right)!$ combinations.

The Shapley value of an eigenvalue is not the difference between the predicted values after removing the feature from the model training. Given the current set of eigenvalues, the contribution of the eigenvalues to the difference between the actual predicted value and the average predicted value is the estimated Shapley value. For the tree model used in this paper, the algorithm pseudo-code for calculating the Shapley value can be expressed in the following Algorithm 1.

In Algorithm 1, ***v*** is a vector of the node values, the vectors ***a*** and ***b*** are the left and right node indices of each internal node, the vector t is the threshold for each internal node, and ***d*** is the index vector of the features split between the internal nodes. The vector ***r*** represents the number of samples covered by each node, which represents how many data samples fall into that node. The weight ***w*** represents the proportion of training samples that match the subset *S* that fall into the nodes. 

**Algorithm 1:** SHAP Tree Model Shapley Value Calculation Pseudo-Code $E\left[f\left(x\right)|x_{S}\right]$.**Inputs: *x***,***S***,***tree***={***v***,***a***,***b***,***t***,***r***,***d***}
**Process:**
1: if vj does not belong to an internal node, then2:  return *w**vj
3: elseif 4:  if dj∈S, then5:    if xdj≤tj
6:      return Gaj,w
7:    else return Gbj,w
9:    endif8:  else return Gaj,wraj/rj+Gbj,wrbj/rj
9:   endif10: endif**Output:** Gj,w

### 2.3. Causal Inference

The correlation described in the contribution section of the SHAP model is the basis of the “prediction”, while the causal relationship is the basis of the “decision”. Most existing machine learning algorithms are association-driven, which leads to their erratic performance when applied to test data, because the distribution of the test data may be different from that of the training data. The causality is assumed to be invariant in the datasets. A reasonable approach is to explore causal knowledge so as to achieve interpretable stable predictions.

For causality, randomized experiments are the golden rule. However, in practical scenarios, the A/B test is often not feasible due to various factors, such as ethics, costs, and technology. In industrial systems, it is very expensive to conduct random control experiments, especially in the field of rail transit. Such a procedure usually collects data at the expense of online efficiency. Usually, we are not allowed to conduct randomized controlled experiments. In this context, in this paper, we conduct a study of the causal effects on an observational dataset that was collected.

This chapter describes the method for measuring causal effects based on observational data. The method is divided into 4 steps: causal graph construction, identification, estimation, and refutation.

#### 2.3.1. Causal Graph Construction

We create an underlying causal graph model for the problem under study. The graph does not have to be complete. A local graph can be proposed to represent the prior knowledge about some variables. The model treats the remaining variables as potential confounders in the subsequent calculation steps.

In this paper, a directed acyclic graph (DAG) is used to characterize the causal relationships between variables. As shown in Figure 5, the nodes represent the variables or features in our research question. Without limitation, let us understand each node as representing something that is potentially observable, measurable, or otherwise comprehensible for the system. Edges connect the nodes to each other. Each edge represents a mechanism or causal relationship related to the value of the connected nodes. The edges are used to indicate the flow of causal influences. Define the variable *T* as the treatment, *Y* as the outcome, and *W* as the covariate. The node *T* arrow points to *W*, which means that *T* is the cause of *W*. Call the *T* node the Ancestor and the *W* node the Descendant. There are three basic structures of a directed acyclic graph, namely the chain structure, fork structure, and collider structure.

Chain structure:

Common in the front door path, *T* affects *Y*, which must pass through *W*.

Fork structure:

The intermediate node *B* is usually regarded as the common cause or confounder of A and C.

The confounding factor *W* will cause *T* and *Y* to be statistically associated, even though they are not directly related.

Classic example: “Shoe size ← child’s age → reading ability”, i.e., children who wear larger shoes may be older; thus, they tend to have a stronger reading ability. However, when the age is fixed, *T* and *Y* are conditionally independent.

Colliding structure:

*T* and *W*, as well as *W* and *Y*, are correlated, whereas *T* and *W* are not correlated. Classic example: “beauty → star ← acting”, i.e., when we know a person is a star and she does not have very good acting skills, usually she will be beautiful. Given *W*, *T* and *Y* shift from uncorrelated to correlated.

#### 2.3.2. Identification

After assuming a causal relationship through a causal graph model, we can analyze the causal relationship between the features, as well as the observed features. In this way, we can determine whether there is enough information to answer a particular causal inference question. Identification is the process of analyzing our model. To identify whether *T* is the cause of *Y*, a simple method is to intervene in *T* and see if *Y* changes. The expression is: P(*Y*|Do(*T*)). Randomized experiments are considered the gold standard for identifying causal effects, in which P(*Y*|Do(*T*)) is equivalent to P(*Y*|*T*). The average causal effect (ACE) can be used to measure the effect of an operation. For example, in order to determine the effectiveness of technical improvements, assuming that the intervention operation aims to improve the entire train or not, we then compare the operational health values under the two interventions. We use *do*(*X* = 1) to indicate that all the trains are technically improved and *do*(*X* = 0) to indicate that not all the trains are to be technically improved, and the difference between the two is the average causal effect ACE. The expression is as follows:(5)$$ACE=P\left(Y=1|do\left(T=1\right)\right)-P\left(Y=1|do\left(T=0\right)\right),$$
where $P\left(Y=1|do\left(T=1\right)\right)$ represents the probability of normal operation under the technical improvement intervention, and $P\left(Y=1|do\left(T=0\right)\right)$ represents the probability of the normal operation of the train without technical improvement intervention.

Considering the safety and cost of the research object in this paper, it is difficult to conduct a randomized trial. Based on observational data, this paper transforms intervene expressions into expressions involving only observations through do-calculus rules. This process is called identification.

The do-calculus contains 3 basic rules, where *T* is the treatment, *Y* is the outcome, and *Z* refers to the relevant covariates in the following expressions.

Rule 1—Insertion or deletion of observations: If we observe that the variable *W* is independent of the outcome *Y*, then the probability distribution of *Y* will not change with it, and its expression is:(6)$$P\left(Y|do\left(T\right),Z,W\right)=P\left(Y|do\left(T\right),Z\right).$$

Rule 2—Action/observation exchange: If the variable set *Z* blocks all the backdoor paths from *T* to *Y*, then with *Z* as the condition, the intervention data *do*(*T*) is equivalent to the observation data observe(*T*), and its expression is:(7)$$P\left(Y|do\left(T\right),Z\right)=P\left(Y|T,Z\right).$$

Rule 3—Insertion/deletion of actions: If there is no causal path from *T* to *Y*, we can remove *do*(*T*) from the expression, which is:(8)$$P\left(Y|do\left(T\right),do\left(Z\right),w\right)=P\left(Y|\left(T\right),w\right).$$

Based on the above three basic rules of the *do*-operator, if the structure of the entire DAG is known and all the variables are observable, then we can calculate the causal effect between any set of variables according to the above formula of the *do*-operator. However, in the vast majority of practical problems, we neither know the structure of the entire DAG nor are able to observe all the variables. Therefore, simply applying the above formula is not enough. Based on the basic rules of the *do*-operator, we can derive the adjustment formula and the front door criterion. The meaning is as follows:(1)In some studies, even if some variables in the DAG are unobservable, we can still estimate some causal effects from the observed data.(2)These two criteria help us to identify “confounding variables” and design observational studies.Adjust the formula:

As shown in Figure 6, T is the treatment, *Y* is the outcome, and *W* and *Z* are associated covariates. We can consider a set of ordered variables (*T*,*Y*) in a directed acyclic graph. When there is no descendant node of *T* in *W*, and *W* blocks every direct path between *T* and *Y* that contains an edge pointing to *T*, then *W* satisfies the conditions for using the following adjustment formula for (*T*, *Y*). Causal inferences can be determined from observational data.
(9)$$P(Y=y|\mathrm{do}\left(T=t\right))=\underset{z}{\sum }P\left(Y=y|T=t,W=w\right)P\left(W=w\right),$$
where $P(Y=y|\mathrm{do}\left(T=t\right))$ is the probability of the result that *Y* = y after applying the intervention *T* = t. $\sum_{z}P\left(Y=y|T=t,W=w\right)P\left(W=w\right)$ is the probability sum of the result *Y* = y under different observational data *Z* when the *W* covariate is given.

Front Door Guidelines:

As shown in Figure 7, there is an unobservable confounding factor *U* that is a common cause of *T* and *Y*. Since there is no statistical information for *U*, an adjustment formula cannot be used to estimate the causal effect of *T* on *Y*. However, if there is an additional observable variable *Z* lying between *T* and *Y* as a mediator, the causal effect of *T* on *Y* is identifiable in this case. This situation satisfies the front door criterion and can be calculated using the front door criterion formula.

The front door criterion meets the following conditions:*Z* cuts off all the direct paths from *T* to *Y*There is no backdoor path from *T* to *Z*All *Z*-to-*Y* backdoor paths are blocked by *T*

The set of variables *Z* is deemed to satisfy the front door criterion for the pair of ordered variables (*T*, *Y*). The causal effect of *T* on *Y* is identifiable when *Z* satisfies the front-door criterion for the pair of ordinal variables (*T*, *Y*) and P(*T*, *Z*) > 0. It is calculated as follows:(10)$$P\left(Y=y|do\left(x\right)\right)=\underset{z}{\sum }P\left(z|x\right)\underset{x^{\prime }}{\sum }P\left(y|x^{\prime },z\right)P\left(x^{\prime }\right),$$
where $P\left(Y=y|do\left(x\right)\right)$ is the probability under the intervention, and $\sum_{z}P\left(z|x\right)\sum_{x^{\prime }}P\left(y|x^{\prime },z\right)P\left(x^{\prime }\right)$ is the observation probability under the condition of the front door criterion.

In some cases, we may not be able to identify an effect based on the model and available data. In such cases, we may reconsider the modeling assumptions, collect new types of data, or state that it is impossible to identify causal relationships.

#### 2.3.3. Estimate

Identification is the process of converting causal quantities into statistics, which are called identification estimators. Once these are identified, estimation is the process of calculating that quantity using the available data. Evaluation is the process of analyzing the data. Commonly used estimation methods include linear regression, stratification, matching, propensity score methods, weight-based methods, etc.

This paper uses linear regression and propensity score methods to estimate different scenarios. Linear regression is a basic inductive statistical method and is not described in this article. The core idea of the propensity score method is that it is difficult to judge whether two samples are “completely matched” when the dimension of the confounding variable is high. We quantify whether a sample matches by setting a score. We use a threshold for the score to define an “exact match”. At the same time, by controlling this score, calculated by X, the experiment is guaranteed to be random. The expression is:(11)$$T⊥X|e\left(X\right),$$
where T is the result, ⊥ means that the variables on both sides of the sign are independent, and $X|e\left(X\right)$ is the variable matching the quantification.

This score can be expressed as the probability of applying treatment (*T* = 1) to the sample given *X*, which is called the propensity score. The expression is:(12)$$e\left(X\right)=P\left(T=1|X\right).$$

In this paper, the Mahalanobis distance is used as the basis for calculating the matching degree of the samples. It accounts for unit differences by normalizing each dimension by the standard deviation. Its expression is:(13)$$Mahalanobis\left(\vec{x_{i}},\;\vec{x_{j}}\right)=\sqrt{{\left(\vec{x_{i}}-\vec{x_{j}}\right)}^{T}S^{-1}\left(\vec{x_{i}}-\vec{x_{j}}\right)}\;,$$
where $\left(\vec{x_{i}},\;\vec{x_{j}}\right)$ refers to the two samples used for calculating the Mahalanobis distance, and S is the covariance matrix.

The difference between the propensity scores is reflected in the distance, and the similarity between the samples in the exact match problem can be measured by the propensity score. Setting a threshold ϵ, if the propensity score of the multi-dimensional feature of the sample is less than the threshold, the sample is regarded as matching. The expression is:(14)$$\mathrm{Distance}\;\left(X_{i},X_{j}\right)=\left|e\left(X_{i}\right)-e\left(X_{j}\right)\right|\leq \epsilon \;.$$

With the above sample propensity scores, we can estimate the causality through the average treatment effect (ATE) and conditional average treatment effect (CATE) without having to consider the effects of confounding variables, because propensity scores enable the treatment and control groups to be within similar propensity score intervals to approximate randomized trials. The expression for the ATE and CATE is:(15)$$\mathrm{ATE}≔E\left[Y|do\left(T=1\right)\right]-E\left[Y|do\left(T=0\right)\right]\;\to E\left[Y|T=1\right]-E\left[Y|T=0\right]\to \frac{\sum_{i=1}^{N}1_{\left[T=1\right]}Y}{\sum_{i=1}^{N}1_{\left[T=1\right]}}-\frac{\sum_{i=1}^{N}1_{\left[T=0\right]}Y}{\sum_{i=1}^{N}1_{\left[T=0\right]}};$$
where $E\left[Y|do\left(T=1\right)\right]$ is the expectation of the result *Y* when the *T* = 1 intervention is applied, and $E\left[Y|T=1\right]$ is the expectation of the result *Y* under the observational data *T* = 1.
(16)$$\mathrm{CATE}≔E\left[Y|do\left(T=1\right),C=c\right]-E\left[Y|do\left(T=0\right),C=c\right]\;\;\;\;\;\;\;\to E\left[Y|T=1,C=c\right]-E\left[Y|T=0,C=c\right]\;\;\;\;\;\;\;\to \frac{\sum_{i=1}^{N}1_{\left[T=1,C=c\right]}Y}{\sum_{i=1}^{N}1_{\left[T=1,C=c\right]}}-\frac{\sum_{i=1}^{N}1_{\left[T=0,C=c\right]}Y}{\sum_{i=1}^{N}1_{\left[T=0,C=c\right]}}\;.$$
where $E\left[Y|do\left(T=1\right),C=c\right]$ is the expectation of the result *Y* when the *T* = 1 intervention is applied under the given condition *C* = c, and $E\left[Y|T=1,C=c\right]$ is the expectation of the result *Y* under the observed data *T* = 1 under the given condition *C* = c.

#### 2.3.4. Refute

After the causal diagram is drawn, the model is determined by identification, and the model is verified by estimation. We can verify the validity of the model by means of refutation. The refutes are tested using different data interventions to verify the validity of the causal effects. The basic principle of refute is that after some intervention based on the original data, the causal effect is re-estimated based on the new data.

In theory, if there is, indeed, a causal effect between the treatment variable (Treatment) and the outcome variable (Outcome), then this causal relationship will not change with changes in the environment or data. The new causal effect estimates are not greatly different from the original estimates.

There are five commonly used countermeasures, as follows below.

Adding random common causes: After we add independent random variables to the dataset as common causes, the causal effect from the estimation method should not change.

Placebo treatment: When we replace the true treatment variable with an independent random variable, the estimated causal effect is zeroed out.

Dummy outcomes: When we replace the true outcome variable with an independent random variable, the estimated causal effect is zeroed out.

Simulation results: When we replace the dataset with a simulated dataset based on the known data generation process closest to the given dataset, it should match the effect parameters in the data generation process.

Data subset validation: When we replace a given dataset with a randomly selected subset, the estimated effect should not change significantly.

In this paper, due to the different positions of the treatment variables (Treatment) in the causal diagram and the data distribution of the variables themselves, different refutation methods are used to verify the causal inference results.

### 2.4. Model Training and Prediction

During the data preparation phase, feature ranges are selected using the prior knowledge of product experts. In the feature contribution calculation stage, the feature contribution is calculated using the SHAP model. In the causal inference stage, the dependent variable that causes the failure is obtained through the causal inference diagram and verified by the method of rebuttal. In the model training and prediction stage, we introduce the dependent variable time series data to improve the influence of this feature on the model prediction, while filtering the confounding features and features with low contributions from the model input. The key information of the fault diagnosis and prediction may be hidden in the time series data of the dependent variable; thus, the introduction of the time series data of the dependent variable can improve the accuracy of the model prediction. The existence of confusing features does not aid in the fault diagnosis and prediction of the model, but it does lead to the overfitting of the model and reduces the generalization ability of the model. Deleting confusing features and features with low contributions from the model input data can reduce the computational complexity of the model and improve the operating efficiency of the model. The specific process of the model training and prediction is as follows.

Introduce the feature time series data: We select the feature values with high feature contributions and causal relationships with the faults based on the feature contribution, causal inference, and expert experience. On this basis, this paper proposes a method that can be used to incorporate the time series data of important features into the dataset in order to improve the effectiveness of the model. The method incorporates one or more samples of the selected feature historical sampling interval into the dataset as new features.

Dividing the dataset: In practice, the proportion of fault data is much smaller than the proportion of normal data. In order to improve the effectiveness of the model, when dividing the dataset, it is necessary to keep the proportions of fault samples and normal samples at a considerable level.

Model training: This paper uses the classic Random Forest, K-NearestNeighbor (KNN), Naive Bayes, Gradient Boosting Decision Tree (GBDT), and eXtreme Gradient Boosting (XGBoost) as the comparison baselines and compares the model prediction effects before and after the method is adopted.

Hyperparameter optimization: Since the method of introducing feature time series data can easily cause the overfitting of the model, the hyperparameter selection of the model selects 3 layers or less and the number of iterations as less than 30 times.

## 3. Results

In order to more clearly explain the results of the feature contribution, described in Section 3.2, and the causal inference results, described in Section 3.3, the dataset is described in Section 3.1.

### 3.1. Dataset Description

The fault of the communication electronic board of the traction converter is the research object of this paper. The failure samples in the dataset are labeled. The dataset in this paper is derived from six types of models and 96 trains of various major high-speed railway lines in the country. Considering the faults of the traction converter electronic boards studied in this paper, 88 were preliminarily screened out from more than 2000 eigenvalues based on the prior knowledge of the experts. The total number of samples in the dataset is 46,000. It includes physical quantity data collected based on sensors, status data based on the train network system, operational data based on the railway passenger transport system, and early warning and judgment data based on the electronic board of the traction equipment of the high-speed EMU. The details are described in Table 1.

### 3.2. Feature Contribution Results

The feature contribution degree is calculated by the SHAP model. The visualization effect is shown in Figure 8. The X-axis is the contribution of each feature to the prediction result. The positive X-axis demonstrates that the feature has a positive contribution to the prediction result, and the negative X-axis demonstrates that the feature has a negative effect on the prediction result. The feature contribution of each feature is represented by the Y-axis. The physical quantity categories are sorted according to the feature contribution, including the power feature, voltage feature, temperature feature, state feature, and pressure feature. The red and blue bars on the right represent the magnitude of the eigenvalues. Each color point is a sample of a specific feature. For example, a low intermediate voltage value has a positive contribution to the fault prediction of the electronic board of the traction equipment, and a high speed per hour has a positive contribution to the fault prediction.

By calculating and visualizing the contributions of the SHAP features, data experts can provide interpretable and intuitive references to the product experts.

### 3.3. Causal Inference Results

The feature contribution obtained by the SHAP model is the correlation of the data, and the correlation can be used as the basis for the fault diagnosis or early warning. However, the facts are often complicated. The fact that the statistics are correlated does not mean that two events are causally connected. More precisely, correlation does not directly imply causation. The causal inference method described in this paper can verify the causality based on highly correlated features.

Local mutation at the moment of failure and global variance can help product experts to gain a better understanding of characteristic data. Features with high contributions are selected, and their potential causal relationships with the faults are preferentially analyzed.

First, we construct a causal graph with the high-contribution features. The average treatment effect is then calculated by the causal inference method described in this paper. Finally, the correctness of the causal diagram is verified by the method of causal inference and refutation. The causal inference experimental data table is shown in Table 3. A cause-and-effect diagram based on the data sheet and prior knowledge of the product experts is shown in Figure 9. Some high-contribution features are verified in the calculation process of causal inference with relatively low ATE values, such as the auxiliary converter output current.

Product experts can use feature contribution and causal inference ATE values as the basis for decision making so as to optimize and improve the products.

### 3.4. Model Prediction Results

Based on the SHAP feature contribution degree and causal inference graph, the features with high contribution degrees and causal relationships with the faults are selected in order to introduce their time series data as a new feature dimension. The further mining of the eigenvalues may hide the fault diagnosis and prediction value in the previous sequence. At the same time, the features that do not have a causal relationship are removed to reduce the data dimension and possible negative impacts on the fault diagnosis and prediction.

The accuracy, precision, recall, and F1 value are used to comprehensively evaluate the effect of this model and conduct ablation experiments. The mathematical expressions of each evaluation method are as follows:(17)$$\mathrm{Accuracy}\;=\frac{TP+TN}{TP+TN+FP+FN}\;,$$
(18)$$\mathrm{Precision}\;=\frac{TP}{TP+FP}\;,$$
(19)$$\mathrm{Recall}\;=\frac{TP}{TP+FN}\;,$$
(20)$$F1=\frac{2\times \mathrm{Precsion}\times \mathrm{Recall}}{\mathrm{Precsion}+\mathrm{Recall}}\;,$$

*TP* is the true positive example, *TN* is the true negative example, *FP* is the false positive example, and *FN* is the false negative example.

The model results of the ablation experiments are shown in Table 4.

“Original” means that the features of the current moment are used as the input for the classical model.

“Input time series” means that the features are not filtered by feature contribution and feature causality, but local time series data of all the features are introduced into the dataset as a new data dimension.

“Causal Inference” is the method proposed in this paper, which introduces high-contribution and causal feature time series data and deletes low-contribution features.

The bold font represents the optimal value of each model under each method.

It can be seen from the results that:(1)Through the causal inference method proposed in this paper, compared with the classical model of the original input, the failure prediction of each model achieves an average improvement of 10% in the accuracy, an average of 21% in the precision, an average improvement of 23% in the recall rate, and an average improvement of 24% in the F1 value.(2)Comparing the “Input time series” method and the “Causal inference” method proposed in this paper, the introduction of the full-feature time series data method increases the feature dimension complexity to 270%. The calculation time is 35% longer than that of the method proposed in this paper.(3)The method of applying full time series data increases the computational complexity. However, the failure prediction effect is equal or inferior to that of the method proposed in this paper.(4)The causal inference method proposed in this paper can achieve results using various machine learning models and has a good generalization ability and robustness.

The calculation time includes the data import, model building, data fitting, and result display. All the data experiments were conducted using computing infrastructure, as follows: Intel(R) Core(TM) i7-8550U CPU @ 1.80 GHz, 8 GB DDR4 RAM, 256 GB SSD.

## 4. Discussion

This chapter discusses why the method proposed in this paper is effective, the scenarios in which it can be applied, the limitations of the method, and future prospects.

### 4.1. Discussion of the Model’s Validity

Based on the changeable working conditions of products in the rail transit industry and the complex correlations between the characteristics of each system, identifying suitable sensor characteristics is still the key to fault diagnosis and prediction. By selecting the correct features through the method of feature engineering described in this paper, the machine learning model can achieve a higher accuracy and computational efficiency. The method proposed in this paper can be effective in the context of the research object, mainly for the following reasons:As described in Section 2.1, during the data preparation, product experts conduct preliminary feature screening based on prior knowledge, which greatly reduces the data dimension and the complexity of the subsequent model calculations.As described in Section 2.2, during the calculation of the feature contribution degree, the data experts calculate the correlation between each feature and the fault using the SHAP model. The feature dimension is further reduced to 26, and the visual SHAP value contribution graph of the correlation ranking of each feature is provided.As described in Section 2.3, in the causal inference stage, the product experts draw causal inference diagrams based on prior knowledge. Through the steps of identification, estimation, and refutation, the feature causal relationship decision recommendation is obtained according to the data. The product experts then re-examine and modify the causal inference graph based on prior knowledge and, finally, obtain eight important features with causal relationships.In the model training and prediction link stage, as described in Section 2.4, the diagnostic and prediction capabilities of the model are provided by introducing characteristic time series data with high contributions and causal relationships. At the same time, the feature data with low contributions and no causal relationships are deleted to avoid the overfitting of the model and reduce the robustness of the model.

### 4.2. Application Scenarios

It is necessary to solve the challenges related to the robustness, accuracy, and interpretability of the high-speed train fault early warning. Based on the early warning data for the on-site high-speed train traction equipment electronic board, the method proposed in this paper is rendered more robust, accurate, and interpretable through ablation experiments. The method proposed in this paper can be transferred and applied to other research objects in the context of fault warning and the product optimization of other key components of high-speed railway, as follows:Diagnostic warning based on the model’s robustness and accuracy:

In application scenarios where there is no principle-driven fault early warning mechanism, and the diagnosis cost is high and the misjudgment cost is low, the method can provide indirect diagnosis–risk early warning of faults in a data-driven manner.

For application scenarios involving an existing principle-driven fault early warning mechanism where the diagnosis false alarm rate is high and the diagnosis equipment failure rate is high, the method can form a joint diagnosis with the existing diagnosis method in a data-driven manner.

Model Visualization-Based Product Optimization:

Compared with fault diagnosis and early warning, the optimization of the performance and quality of products to reduce the probability of failure can better protect the safety of the passengers and property and the travel efficiency. However, under the influences of complex working conditions and a massive amount of data, product optimization performed in a principle-driven manner has high trial-and-error costs and time costs. If it is supplemented by the model feature contribution and interpretability method proposed in this paper, it can provide data support for technical experts. For example, experts can deduce which feature should be considered first and which device should be used first in an attempt to optimize the data support. With the support of experts, using the method proposed in this paper, the efficiency of the product optimization can be improved.

In Figure 10, the meanings of the coordinate axes are the same as those in Figure 8. The color scale bar at the top is the final early warning result of the sample after the SHAP contribution of each feature has been added. The closer the line in the figure is to 1, the higher our confidence is that the warning is a fault. The closer the line is to 0, the higher our confidence is that the prediction is normal. The dotted line indicates that the predicted value of this sample does not match the true value.

In Figure 11, the bold numbers on the coordinate axis represent the prediction result and its confidence in regard to the sample. The closer it is to 1, the higher our confidence is that the warning is a fault. The closer the line is to 0, the higher our confidence is that the prediction is normal. The red and blue color blocks on the X-axis represent the cumulative SHAP contribution of each feature. Red represents a positive contribution to the prediction of failure, and a negative contribution to the prediction of a normal outcome. Blue represents a positive contribution to the prediction of a normal outcome and a negative contribution to the prediction of a fault.

As shown in Figure 9, through the steps of causal inference, identification, estimation, and refutation, product experts can obtain decision suggestions generated by causal inference in the process of product optimization. The method of causal inference provides product experts with a tool that can be used to input prior knowledge. Based on the reinforcement of prior knowledge, the method of causal inference can provide decision support for product experts in order to identify the cause of the failure.

Product optimization experts can visualize the contribution of each feature to the failure through the process depicted in Figure 8 and Figure 10. The process in Figure 9 removes features that have an incorrect causal relationship with the failure. The product experts can determine the priority for the product’s improvement based on the level of the feature contribution and causal inference, combined with physical principles.

Fault warning experts can study and analyze the misjudged single-sample model decision trajectory and its related characteristics through the process depicted Figure 10 and then optimize the accuracy of the model.

### 4.3. Disadvantages and Prospects

Disadvantages

There are two disadvantages when the method proposed in this paper is employed in industrial applications:(1)There are usually superficial causes and root causes of the failures in field industry applications. The causal inference model proposed in this paper does not provide feature causal chain decision support. Product experts may identify the shallow cause of the failure, but they may not be able to identify the root cause of the failure.(2)Selecting features with high contributions and introducing them into the time series data improves the model’s effect. However, the new data introduced into the model algorithm is used as another feature for the training. Data from different time series for the same feature are not related in the model.Prospects

We can introduce multi-dimensional enterprise operational data in design, manufacturing, and after-sale scenarios as eigenvalues to further improve the accuracy of the model. The dimension of the eigenvalues is enriched by adding data, such as engineering change data in the design process, one-time inspection pass rate data in the manufacturing process, and failure repair rate data in the after-sale process, so that the model has a higher robustness, accuracy, and interpretability.

## 5. Conclusions

This paper proposes a machine learning model for high-speed train fault early warning based on feature contribution and causal inference. Firstly, in the data preparation process, the data collected from the train sensors is processed for the balancing of positive and negative samples, missing values, abnormal sample removal, and normalization. Additionally, one can select features based on prior knowledge to initially reduce the dimension of the features. Secondly, in the feature contribution calculation process, the feature contribution is calculated using the SHAP model. We then plot the SHAP feature contribution graph. Features with high contributions are obtained, which further reduces the dimension of the features and the complexity of the subsequent model calculations. Thirdly, in the causal inference process, through the steps of the creation of a causal diagram, identification, estimation, and refutation, and by combining the prior knowledge of product experts to identify the features that have causal relationships with the fault, the dimension of the features is reduced again. Finally, in the model training process, the model training and prediction are carried out by introducing time series data on features with high contributions and causal relationships.

Taking the high-speed train traction converter electronic board failure dataset as an example, the method proposed in this paper can be used to filter features, introduce feature time series data with high contributions and causal relationships, and delete low contribution features. The optimal prediction effect is achieved with the lowest computational complexity. Ablation experiments are performed based on the classic machine learning models KNN, XGBoost, Random Forest, GDBT, Naïve Bayes, and Logistic Regression. The results show that after the optimization of the method proposed in this paper, the fault diagnosis and prediction achieve an average improvement of 10% in the accuracy, precision, recall, and F1 value. At the same time, the computational complexity is reduced by 63%. The causal inference graph derived from the method in this paper can provide product experts with visual and interpretable decision support during product optimization.

A case is formed in industry applications through the method proposed in this paper. The case shows that product experts and data experts can jointly optimize a product’s performance and improve model warning effects through cooperation. The future research directions will be to optimize the causal inference model, obtain the fault propagation chain and causal feature chain, establish associations between time series data in regard to the same feature, and introduce multi-dimensional enterprise operational data in design, manufacturing, and after-sale scenarios as eigenvalues so as to further improve the accuracy of the model.

## Figures and Tables

**Figure 1 sensors-22-09184-f001:**
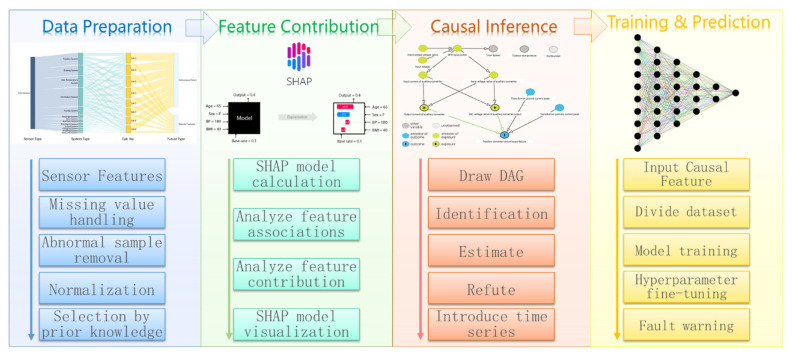
Overview of the research methods.

**Figure 2 sensors-22-09184-f002:**
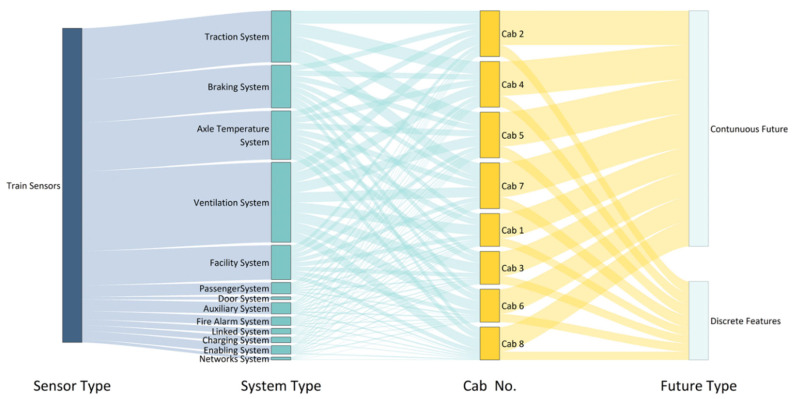
Alluvial diagram of the sensor data distribution.

**Figure 3 sensors-22-09184-f003:**
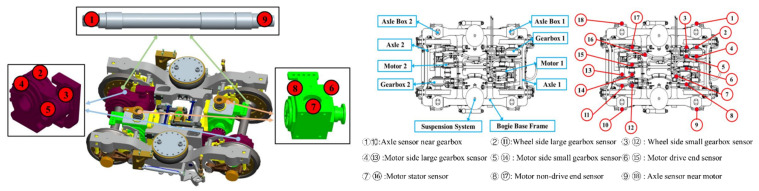
Schematic diagram of high-speed train sensor distribution.

**Figure 4 sensors-22-09184-f004:**
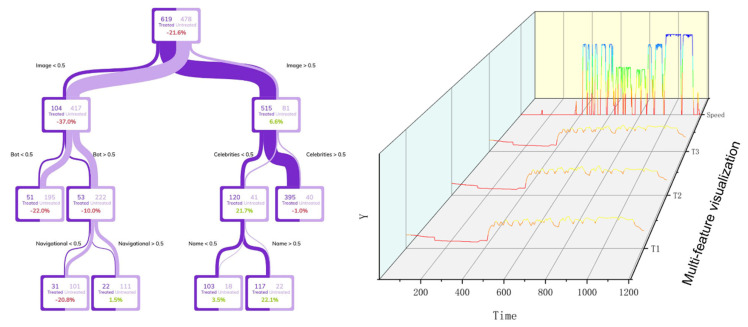
Decision tree visualization and data 3D waterfall diagram illustration.

**Figure 5 sensors-22-09184-f005:**
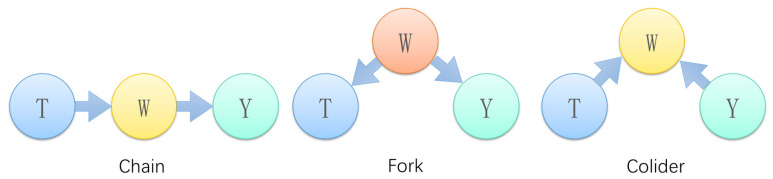
Diagram of three basic structures of a directed acyclic graph.

**Figure 6 sensors-22-09184-f006:**
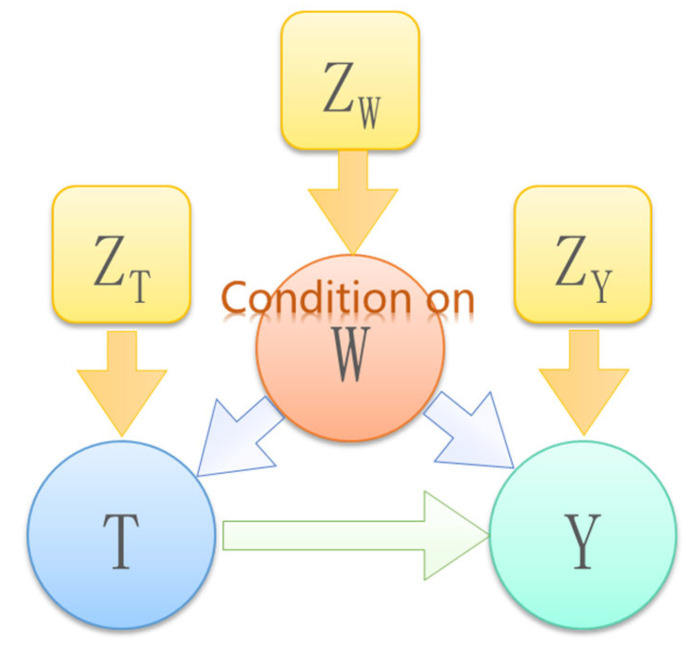
Adjustment formula application DAG.

**Figure 7 sensors-22-09184-f007:**
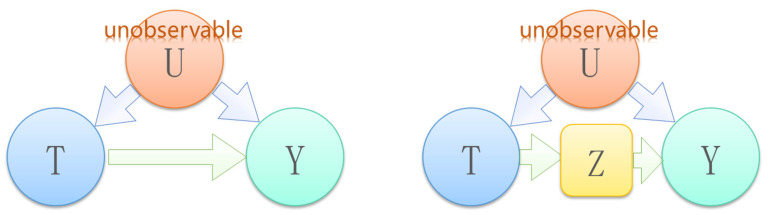
Front door criterion application DAG.

**Figure 8 sensors-22-09184-f008:**
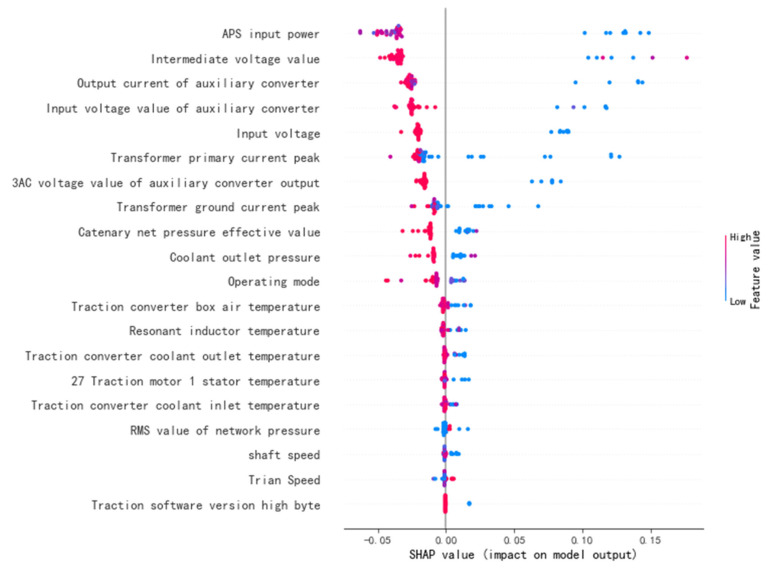
A visualization of the SHAP value of each feature contribution.

**Figure 9 sensors-22-09184-f009:**
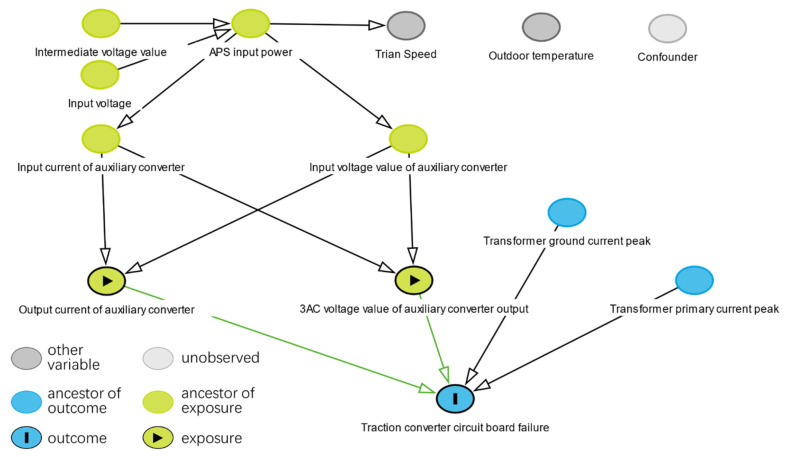
Feature causal inference graph of traction converter circuit board failure.

**Figure 10 sensors-22-09184-f010:**
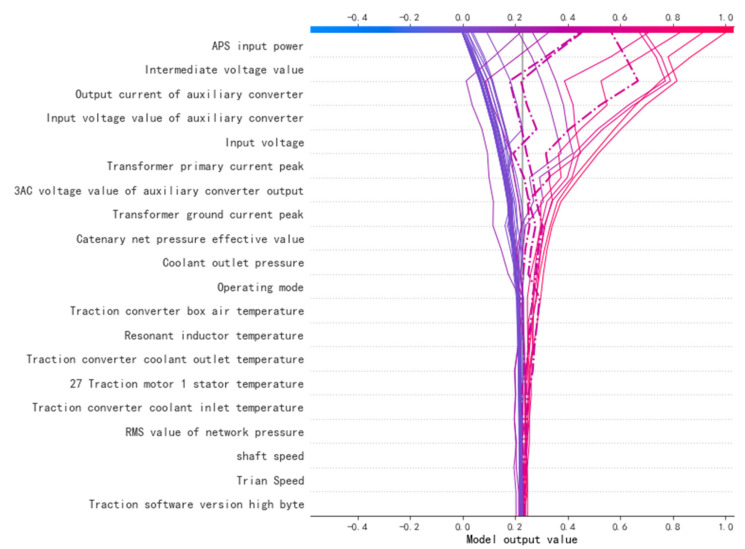
Visual illustration of the decision path of each sample based on the SHAP value of each feature contribution.

**Figure 11 sensors-22-09184-f011:**
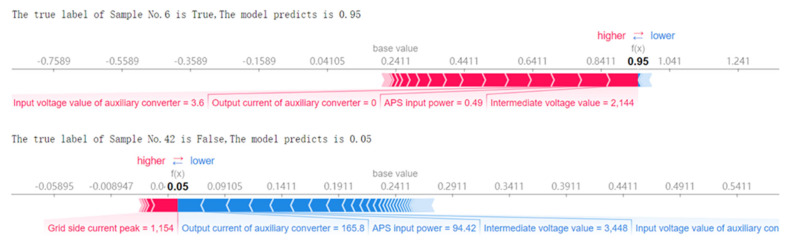
Visual illustration of the influence of each feature contribution’s SHAP value on the single-sample prediction.

**Table 1 sensors-22-09184-t001:** Dataset type description and examples.

Data Type	Data Category	Units/Format	Data Example
Sensor data	Speed	km/h	Train speed, shaft speed, tug shaft speed, etc.
	Temperature	°C	Outdoor temperature, traction motor stator temperature, traction converter coolant temperature, etc.
	Pressure	Pa	Coolant inlet/outlet pressure, air spring pressure, etc.
	Voltage	V	Intermediate voltage value, input voltage, catenary voltage, etc.
	Current	A	Auxiliary converter output current, transformer primary current peak value.
	Power	W	APS input power, traction converter inverter AC side power, etc.
	Weight	kg	The actual quality of the train, etc.
Status data	Time	YYYY-MM-DD hh:mm:ss	Sample sampling time.
	Operating state	1/0	Train maneuver command status.
	Diagnostic status	#	Train life signal, fault code, etc.
Operational data	Train information	#	Line, train number, car number, etc.
Early warning data	Judgment results	1/0	Fault occurrence, fault removal, etc.

**Table 2 sensors-22-09184-t002:** Various data characteristics, examples, and preprocessing methods.

Type	Characteristic	Mathematical Meaning	Average Representation	Example	Preprocessing Method
Classify	Mutually exclusive separable classes	=, ≠	Mode	Car model	One-hot encoding
Sequencing	Rank order	>, <	Median	Level 1–Level 3	One-hot encoding
Distance	Continuous value, no multiples	+, −	Arithmetic mean	Celsius	Normalized
Fixed ratio	With 0 points, multiples comparable	+, −, ×, ÷	Geometric mean	Voltage and current	Normalized

**Table 3 sensors-22-09184-t003:** Causal inference experimental data sheet.

Feature	Local Mutation	Global Variance	Max SHAP Value	ATE	Random Common Causes ATE	Placebo Treatment ATE
Intermediate voltage	3637	1015	0.115	0.46	0.46	0.02
APS input power	98	32	0.096	0.74	0.74	0
Input voltage	384	154	0.069	0.67	0.68	0.01
Input voltage of auxiliary converter	3246	1254	0.078	0.36	0.37	0
Output current of auxiliary converter	98.5	50	0.082	0.08	0.08	0.01
Output voltage of auxiliary converter	380.6	145	0.076	0.93	0.92	0
Transformer ground current peak	188	73	0.054	0.52	0.52	0.01
Transformer primary current peak	189	73	0.071	0.52	0.52	0.01

**Table 4 sensors-22-09184-t004:** Ablation experiment of the model prediction results.

Method	Model	Accuracy	Precision	Recall	F1	Dimension	Computing Time
Original	KNN	0.78	0.61	0.4	0.44	26	5.171
	XGBoost	0.73	0.58	0.51	0.43		
	Random Forest	0.82	0.57	0.4	0.43		
	GDBT	0.73	0.53	0.57	0.48		
	Naïve Bayes	0.84	**0.72**	0.39	0.49		
	Logistic Regression	0.84	0.68	0.61	0.63		
Input time series	KNN	0.86	0.7	0.69	0.67	52	6.008
	XGBoost	**0.92**	**0.88**	0.74	**0.76**		
	Random Forest	0.87	0.85	0.58	0.63		
	GDBT	0.81	0.69	0.57	0.59		
	Naïve Bayes	0.83	0.64	0.42	0.48		
	Logistic Regression	0.87	0.73	0.71	0.71		
Causal inference	KNN	**0.91**	**0.85**	**0.76**	**0.79**	**19**	**4.444**
	XGBoost	0.91	**0.88**	**0.75**	**0.76**		
	Random Forest	**0.89**	**0.86**	**0.68**	**0.7**		
	GDBT	**0.87**	**0.77**	**0.58**	**0.62**		
	Naïve Bayes	**0.87**	0.71	**0.72**	**0.7**		
	Logistic Regression	**0.91**	**0.86**	**0.76**	**0.79**		

## Data Availability

Not applicable.

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
