# Peer review of "Fault Early Warning Model for High-Speed Railway Train Based on Feature Contribution and Causal Inference"

_sensors, 2022, doi:10.3390/s22239184_

Round 1

Reviewer 1 Report

The authors investigated an early warning system for high-speed railway trains. The problem is worth exploring, however, the paper misses most of the dimensions and needs to address. 

The abstract fails to clarify to the new readers what is the problem. how has it been addressed previously? and how this approach is different and efficient. 

Similarly, the abbreviations are used in the abstract. The new readers wouldn't be able to understand these terminologies. 

There are also some typos, such as the sentence "First, the SHAP model is used to calculate the contribution of associated features. then use causal inference models to identify causal linkages." is not clear. 

The paper must specify which equipment failures are notified. 

The authors claimed that they have reduced the computational time, however, they failed to mention the proposed algorithms' computational time. 

The equation needs to be explained in the text. Similarly, most of the terminologies used in the paper are not explained. 

The references need to be updated. Most of the conference references miss the conference location. 

Author Response

Dear reviewer

Thank you for your time and professional review comments.

Kind regards.

Reviewer 2 Report

1. Why choose such machine learning architecture?

2. What is the physical basis of the selected optimal feature?

Author Response

(The authors gave the same response as above.)

Reviewer 3 Report

This paper proposes a fault early warning machine learning model based on feature contribution.

Introduction section can be improved, currently it looks like introduction of thesis, not a research article. Remove all sub-headings from introduction, organize it in paragraphs. 

Add detailed Related Work section after introduction. In the related work section, critically analyse existing/ already done research work. high light limitations, advantages, and provide overview of research gap. 

Atleast add few lines of text between section 2.1 and 2.1.1. 

explain section 2.1.1 in more details, such as dataset template, status information of what? 

section 2.1.2, what is normal operating conditions and what are abnormal conditions? also explain which learning mechanism is used? supervised or un-supervised?

section 2.1.4, samples with data integrity below 80% are regarded as abnormal samples. how this 80% is selected? any rationale behind it? why not 75% or 85%? 

Figure 4 is not clear. 

add few lines of text between section 3 and 3.1. 

Table 3, what are the units used for these parameters? 

add few lines of text between section 4 and 4.1..

Author Response

(The authors gave the same response as above.)

Round 2

Reviewer 3 Report

None